# Exploring CD169^+^ Macrophages as Key Targets for Vaccination and Therapeutic Interventions

**DOI:** 10.3390/vaccines13030330

**Published:** 2025-03-20

**Authors:** Rianne G. Bouma, Aru Z. Wang, Joke M. M. den Haan

**Affiliations:** 1Department of Molecular Cell Biology and Immunology, Amsterdam UMC location Vrije Universiteit Amsterdam, De Boelelaan 1117, 1081 HV Amsterdam, The Netherlands; 2Amsterdam Institute for Immunology and Infectious Diseases, Cancer Immunology, 1081 HV Amsterdam, The Netherlands; 3Cancer Center Amsterdam, Cancer Biology and Immunology, 1081 HV Amsterdam, The Netherlands

**Keywords:** CD169, sialoadhesin, Siglec-1, macrophages, dendritic cells, antigen-presenting cells, vaccines, therapeutic interventions, adaptive immunity

## Abstract

CD169 is a sialic acid-binding immunoglobulin-like lectin (Siglec-1, sialoadhesin) that is expressed by subsets of tissue-resident macrophages and circulating monocytes. This receptor interacts with α2,3-linked Neu5Ac on glycoproteins as well as glycolipids present on the surface of immune cells and pathogens. CD169-expressing macrophages exert tissue-specific homeostatic functions, but they also have opposing effects on the immune response. CD169^+^ macrophages act as a pathogen filter, protect against infectious diseases, and enhance adaptive immunity, but at the same time pathogens also exploit them to enable further dissemination. In cancer, CD169^+^ macrophages in tumor-draining lymph nodes are correlated with better clinical outcomes. In inflammatory diseases, CD169 expression is upregulated on monocytes and on monocyte-derived macrophages and this correlates with the disease state. Given their role in promoting adaptive immunity, CD169^+^ macrophages are currently investigated as targets for vaccination strategies against cancer. In this review, we describe the studies investigating the importance of CD169 and CD169^+^ macrophages in several disease settings and the vaccination strategies currently under investigation.

## 1. General Introduction

CD169 (Siglec-1/Sialoadhesin) is a sialic acid-binding immunoglobulin-like lectin predominantly expressed on specific subsets of tissue-resident macrophages. These specialized macrophages are strategically positioned in various lymphoid and non-lymphoid tissues, including the spleen, lymph nodes, liver, lung, bone marrow, and other organs where they serve as sentinel cells at potential pathogen entry sites and mediate immune responses.

Recent advances have unveiled the multifaceted roles of CD169^+^ macrophages in homeostasis and various pathological conditions. These cells exhibit remarkable characteristics in response to infectious diseases, serving as both protective mediators and potential viral reservoirs. Also, in the context of inflammation and cancer, CD169-expressing cells exhibit multiple functions that can either promote or suppress disease progression, making them attractive therapeutic targets. Furthermore, the unique properties of CD169^+^ macrophages have sparked interest in their potential applications for targeted vaccination strategies and immunotherapeutic approaches.

This review will discuss the current understanding of CD169^+^ macrophages regarding steady-state functions, tissue-specific distribution patterns, and roles in pathological conditions, including infectious diseases, inflammation, and cancer. Additionally, we will explore different CD169-targeting strategies and their applications for vaccine development and therapeutic interventions. Understanding these aspects is crucial for developing novel therapeutic approaches targeting CD169^+^ macrophages in various disease contexts.

## 2. Structure and Binding Function of CD169

CD169 was first discovered as a non-phagocytic sheep erythrocyte receptor on bone marrow macrophages and was subsequently shown to be expressed on splenic marginal zone metallophilic macrophages (MMMs), lymph node subcapsular sinus macrophages (SSMs), but also peripheral tissue macrophages such as liver Kupffer cells and lung alveolar macrophages [1,2,3,4]. CD169 is upregulated on monocytes and macrophages under inflammatory conditions, especially after exposure to type I interferons (IFNs) [5]. Structurally, CD169 is defined as a transmembrane glycoprotein with a long extracellular region [6]. It recognizes α2,3-linked Neu5Ac on both O- and N-linked glycans as well as glycolipids such as gangliosides [2,7,8,9]. CD169 is part of the sialic-acid-binding immunoglobulin-like lectins (Siglec) family, but in contrast to many other Siglecs, it does not have an inhibitory signaling motif [10]. Instead, CD169 functions as an adhesion receptor via *trans* interactions with ligands on other cells or sialylated pathogens [3,10,11,12]. The CD169 glycan binding site has low millimolar affinity for sialylated glycans and, therefore, receptor-ligand clustering needs to take place to enable adhesion [3,13]. It’s unusually long extracellular domain prevents *cis*-inhibition by sialic acids on the cell surface and instead facilitates interaction with ligands on other cells [3].

CD169 presence on macrophages contributes to various processes. Endocytosis of pathogens by CD169^+^ macrophages can occur in a CD169-dependent manner [11,14]. Sialic acids are generally uncommon on the surface of pathogens; however, a number of bacteria have evolved the ability to either synthesize or capture sialic acids and display them on their surface [15]. This effectively blocks the accessibility of other epitopes but allows for CD169-related uptake of pathogens. In addition, enveloped viruses utilize the host membrane which is covered in sialic acids and allows them to bind to macrophages in a CD169-dependent manner [14]. CD169 binding to sialic acids also enhances the interaction of macrophages with other immune cells, such as dendritic cells (DCs), neutrophils, innate lymphocytes, and regulatory T cells [16,17,18,19,20]. The maturation of erythrocytes is supported by CD169-mediated interaction of bone marrow macrophages with erythroblasts [21,22]. To summarize, CD169 is a Siglec with unique properties that enables its participation in many processes including pathogen uptake and immune cell regulation.

## 3. CD169-Expressing Macrophage Subsets Within the Tissues

The function of CD169^+^ macrophage subsets in different organs varies from screening the surrounding fluids and initiating immune responses to supporting other tissue-resident cells (Figure 1 and Table 1). The specific functions of these macrophages per tissue will be described below.

Overview of tissue-resident, CD169-expressing macrophage subsets in lymphoid and non-lymphoid tissues, as described in Section 3. Created in BioRender. den Haan, J. (2025) https://BioRender.com/g25k931 (accessed on 19 February 2025).

### 3.1. CD169^+^ Macrophages in the Bone Marrow

Within the bone marrow, multiple distinct niches exist for the development of erythrocytes, hematopoietic stem cell maintenance, and bone homeostasis. Erythroblastic island niches support the maturation of erythroblasts [21]. Erythroblastic island macrophages express CD169 and aid erythropoiesis through the secretion of cytokines, iron transport and phagocytosis of extruded nuclei [23,24,25,26,27,28]. CD169 on these macrophages interacts with CD43 on early erythroblasts which promotes their maturation [22]. The importance of CD169^+^ erythroblastic island macrophages is further highlighted in a mouse model of *Plasmodium yoelii*. Reportedly, a decrease in CD169 expression as well as a decrease in CD169^+^ macrophages overall is associated with defective erythropoiesis in *P. yoelii*-infected mice [29]. This model additionally proposes a function for CD169^+^ bone marrow macrophages in the defense against incoming parasites by activating plasmacytoid DCs to secrete type I interferons (IFNs) [30,31].

CD169^+^ macrophages in the hematopoietic stem cell niche of the bone marrow are mainly involved in hematopoietic stem cell maintenance and stimulate CXCR4^+^ hematopoietic stem cell retention through support of CXCL12-expressing stromal cells [32]. This is supported by work showing that CD169^+^ macrophage depletion results in retention loss and hematopoietic stem cell mobilization to the blood stream [33]. Moreover, CD169^+^ bone marrow macrophages gain iron through microbiota-induced recycling of red blood cells and use this iron to induce hematopoietic stem cell differentiation [34].

Finally, osteomacs are present within the tissue lining of the bone and express CD169, which aids in distinguishing them from osteoclasts [21,35]. Osteomacs are important inducers of mesenchymal maturation into osteoblasts and additionally support mature osteoblasts in their bone repair functions [36]. Together, these studies indicate that three different macrophage subsets in the bone marrow express CD169 and aid the generation and maintenance of other (immune) cell subsets.

### 3.2. CD169^+^ Macrophages in Spleen and Lymph Nodes

In the secondary lymphoid tissues, spleen, and lymph nodes, specialized populations of macrophages can be identified such as the MMMs and SSMs that express high levels of CD169. Moreover, low expression of CD169 can be found on several other cell subsets, which include splenic red pulp macrophages and marginal zone macrophages, as well as lymph node medullary sinus macrophages. The spleen can be divided into white and red pulp, with the marginal zone functioning as a border between the two. Within the marginal zone of the spleen, the CD169^+^ MMMs are located on top of the B cell follicles. The main function of this macrophage subset is the filtering of the blood in the marginal sinus for antigens and pathogens, a process which is, amongst others, mediated by the CD169 receptor on these cells [37,38,39,40]. A similar cell subset can be identified in the lymph nodes, the SSMs, which are also located on top of the B cell follicles and have access to the subcapsular niche and filter the lymph fluid [41]. Both macrophage subsets depend on B cell-derived lymphotoxin α1β2, as well as liver X receptor and marginal zone reticular cell-mediated receptor activator of NFkB (RANK) signaling [42,43,44,45,46,47,48]. Additionally, lymphatic endothelial cells in the lymph node interact with CD169 through sialoglycans and produce CSF-1 to maintain the SSM population [49,50].

CD169^+^ macrophages can bind intact antigens and retain this on the cell membrane to present to naive B cells [51,52,53,54,55]. Reactivation of memory B cells occurs through scanning of SSMs in their vicinity for antigen [56]. One of the ways in which CD169^+^ macrophages stimulate B cell activation is through the production of IL-18, which is dependent on inflammasome activation [57,58]. In a similar fashion, SSMs are important for the homing and activation of innate-like lymphocytes through interaction with CD169 and IL-18 secretion, respectively [20,59,60]. One of the more recently discovered functions of CD169^+^ macrophages includes the transfer of antigens to cDC1s in order to activate CD8^+^ T cells [16,44,61,62,63,64]. The death of antigen-loaded CD169^+^ macrophages is one of the hypotheses explaining how antigens are transferred to DCs [16]. Interestingly, death of CD169^+^ macrophages has also been linked to persistent chronic infection and disruption of the SSM layer impairs B cell responses in a secondary infection [65,66,67]. Alternatively, acute infection resolution and virus clearance rely on type I IFN production by these macrophages [47,51,65,68,69]. Finally, CD169^+^ macrophages are involved in the suppression of immune responses against self-antigens of apoptotic cells [70,71].

In conclusion, CD169^+^ macrophages in the secondary lymphoid tissues efficiently capture pathogens from blood and lymph fluid and subsequently present these to B cells and transfer them to DCs for T cell priming. This unique feature to retain and present antigens to enhance adaptive immune responses can be utilized to increase vaccine efficacy as is described below.

### 3.3. CD169^+^ Macrophages in the Liver

Kupffer cells are the major population of resident liver macrophages that mediate antimicrobial activity, clear senescent red blood cells and platelets, and contribute to lipid metabolism, tissue repair, and tolerance induction [72,73,74]. A common signature for steady-state human and mouse Kupffer cells that consists of CD5L, VSIG4, CD163, FOLR2, MARCO, and SLC40A1 has been proposed [73]. In addition, CD169 was suggested as one of the useful markers for identifying Kupffer cells, though the expression is lower than on lymphoid tissue macrophages [4,75]. Various liver diseases are characterized by the depletion of resident Kupffer cells and their replacement by bone marrow-derived monocytes that differentiate into Kupffer cell-like cells [73,76,77,78,79]. During cirrhosis, the upregulation of CD169 on monocytes and liver macrophages has been described [80]. Kupffer cell function has been studied using depletion models such as CD169-diphtheria toxin receptor (DTR) transgenic mice or injection with clodronate liposomes. However, these approaches face three key limitations that complicate experimental interpretation. First, the experimental window is constrained because there is rapid replacement by bone marrow-derived monocytes that differentiate into Kupffer cell-like cells [73,76,77,78,79]. Second, macrophage depletion inherently triggers inflammatory responses that can confound results. Third, the CD169-DTR model lacks tissue specificity, causing systemic depletion of CD169^+^ macrophages across multiple organs rather than targeting only liver Kupffer cells. In the context of immune responses, CD169^+^ Kupffer cells not only serve as scavengers for gut-derived pathogens but have also been reported to present antigens to T cells [81,82]. However, this antigen presentation capability remains debatable, as some researchers suggest that the observed phenomenon might actually represent cell doublets between Kupffer cells and liver sinusoidal endothelial cells rather than direct antigen presentation by Kupffer cells themselves [73]. In addition, it is not clear whether Kupffer cells may transfer antigens to DCs as has been described for secondary lymphoid organ CD169^+^ macrophages.

### 3.4. CD169^+^ Macrophages in the Lung

Alveolar macrophages are located in the alveolar lumen and regulate surfactant levels and lipid metabolism as well as providing first-line defense against inhaled pathogens while maintaining an anti-inflammatory environment [83,84,85]. Key surface markers on mouse alveolar macrophages are CD64, MerTK, Siglec-F, CD11c, MARCO, F4/80, and CD169, while CD43, CD169, CD206, CD64, and MARCO are used to identify human alveolar macrophages [83,86,87]. Next to alveolar macrophages that are dependent on GM-CSF [88,89], there are two interstitial macrophage subsets dependent on M-CSF described in the lungs of which one also expresses CD169. However, their functional roles in lung homeostasis and disease remain incompletely understood [83,90,91]. During acute viral infection, alveolar macrophages undergo a phenomenon known as the ‘macrophage disappearance reaction’, similar to what occurs to SSMs during infection [67]. Following this depletion, bone marrow-derived monocytes differentiate into alveolar macrophages to fill the empty tissue niche [83]. CD169 serves as a binding receptor on alveolar macrophages for porcine reproductive and respiratory syndrome virus (PRRSV) and severe acute respiratory syndrome coronavirus 2 (SARS-CoV-2) as discussed below [92,93,94,95]. Interestingly, single nucleotide polymorphisms in Siglec-1 have been associated with active pulmonary tuberculosis and asthma [96,97]. Further research will be necessary to elucidate the role of CD169 on alveolar macrophages in these diseases.

### 3.5. CD169^+^ Macrophages in the Brain, Gut, Skin, and Kidneys

CD169 expression has been reported on different macrophage subsets in multiple other tissues. Brain perivascular macrophages are central nervous system border-associated macrophages and key players in the communication between the brain and the vasculature [98,99]. CD169 expression has been described on these macrophages and is particularly associated with an inflammatory setting [100,101]. In response to brain ischemia, these macrophages accumulate in perivascular locations near the side of injury and portray an inflammatory signature which includes the upregulation of genes associated with neutrophil chemotaxis [100,102].

A vitamin A-dependent subset of CD169^+^ macrophages was identified in the lamina propria of the colon [103,104]. CD169^+^ intestinal macrophages reportedly switch on an acute inflammatory Maf-dependent gene signature after exposure to bacterial components [105]. Additionally, a colitis mouse model showed the importance of CD169^+^ macrophages in the inflammatory response after mucosal damage [104,106]. A resident, CD169^+^ macrophage population with varying functions has also been identified in the skin [107,108]. In the kidney microenvironment, CD169^+^ macrophages regulate neutrophil trafficking upon injury and play an important role in controlling Candida growth via neutrophils [109,110].

While the brain, gut, skin, and kidneys have been described to contain CD169^+^ macrophages, it is not yet clear whether these macrophages share similar features or express unique tissue-dependent functions.

**Table 1 vaccines-13-00330-t001:** CD169-expressing macrophage subsets within the tissues.

Tissue	CD169^+^ Macrophages	Phenotype	Function	References
Bone Marrow	Erythroblastic island macrophages	F4/80^+^ CD169^+^ VCAM1^+^ ER-HR3^+^ Ly6G^+^ Epor^+^	Erythropoiesis support	[24,25]
	Hematopoietic stem cell niche macrophages	F4/80^+^ CD169^+^ CD115^+^ CD68^int^ CD11b^lo^ MHCII^int^ CD11c^int^ CX3CR1^−^	Hematopoietic stem cell maintenance	[33]
	Osteomacs	F4/80^+^ CD169^+^ CD115^+^ CD68^+^ ER-HR3^−^	Bone homeostasis and repair	[21,35]
Spleen	Marginal zone metallophilic macrophages	F4/80^lo^ CD169^++^ MHCII^int^ CD11c^lo^ SIGNR-1^−^ VCAM^+^ ICAM^+^	Pathogen/antigen capture and transfer	[12,37,51]
	Red pulp macrophages	F4/80^++^ CD169^lo^ CD11b^lo^ CD163^+^ CD172a^+^	Erythrocyte clearance and iron recycling	[37]
Lymph node	Subcapsular sinus macrophages	F4/80^lo^ CD169^++^ CD11b^+^ MHCII^+^ CD11c^lo^ VCAM^+^ ICAM^+^	Pathogen/antigen capture and transfer	[45,111]
	Medullary sinus macrophages	F4/80^hi^ CD169^+^ CD11b^+^ MHCII^+^ CD11c^lo^ SIGNR-1^+^	Immune surveillance	[45,111]
Liver	Kupffer cells	CD11b^+^ CLEC4F^+^ CD64^+^ CD169^+^ VSIG4^+^ CD163^+^ FOLR2^+^ MARCO^+^	Blood-borne particle clearance, lipid metabolism modulation	[4,73,75]
Lung	Alveolar macrophages	CD64^+^ CD169^+^ MARCO^+^ MerTK^+^ Siglec-F^+^ CD11c^+^ F4/80^+^	Immune homeostasis and surveillance, surfactant level, and lipid metabolism regulation	[83]
	Interstitial macrophages	CD169^+^ F4/80^+^ MerTK^+^ CD64^+^ MHCII^+^ CD11c^−^	Immune homeostasis	[83,90,91]
Other organs	Perivascular brain macrophages	CX3CR1^lo^ CD169^+^ CD11b^+^ IBa-1^lo^ CD206^+^ CD163^+^ CD64^+^ MerTK^+^	Inflammatory microenvironment regulation	[98]
	Intestinal macrophages	F4/80^+^ CD169^+^ CD11b^+^ MHCII^+^ CD11c^lo^ CX3CR1^+^ CD103^−^ CD135^−^ CD115^+^		[103,104,105]
	Skin macrophages	F4/80^+^ CD169^+^ CD11b^+^ CD64^+^ MerTK^+^		[107,108,112,113]
	Kidney macrophages	F4/80^+^ CD169^+^ CD11b^int^		[110]

## 4. CD169^+^ Macrophages in Disease

Tissue-resident CD169^+^ macrophages have important homeostatic functions as described above. In addition, CD169 is upregulated on monocytes and monocyte-derived macrophages during inflammation. Next, we discuss the role of CD169^+^ macrophages and the CD169 receptor in infectious diseases, auto-immunity, and cancer.

### 4.1. The Role of CD169^+^ Macrophages and the CD169 Receptor in Infectious Diseases

#### 4.1.1. Viral Infection

CD169^+^ macrophages situated at barrier tissues such as the spleen, lymph nodes, liver, and lungs play vital roles in preventing systemic spreading of pathogens. Classical studies using clodronate liposomes and CD169-DTR mice to deplete these macrophages have shown that CD169^+^ macrophages not only behave as fly-paper to contain pathogens but also allow localized production of antigens to activate the immune system as previously reviewed [12,41,114]. In addition, the CD169 receptor itself is described as a receptor for virus capture. This was first described for alveolar macrophages that bind PRRSV in a CD169-dependent manner [95,115]. Secondly, CD169 was shown to be a receptor for binding of enveloped viruses, such as human immunodeficiency virus type-1 (HIV-1), Ebola, as well as murine leukemia virus by binding to gangliosides in the viral membrane [14,116,117,118,119]. For SARS-CoV-2, the binding of CD169 to the spike protein and potentially to gangliosides in the viral membrane was described [120,121,122]. Capture of viruses by CD169^+^ macrophages facilitates subsequent trans-infection of CD4^+^ T cells and B cells, as has been shown for HIV-1, SARS-CoV-2, and murine leukemia virus [116,121,123,124,125]. Once bound to CD169, HIV-1 is internalized into specialized compartments within the DCs known as virus-containing compartments [126,127,128]. The virus can remain infectious within these compartments for extended periods, but this does not lead to infection of the CD169^+^ macrophage and replication. When CD169^+^ macrophages interact with CD4^+^ T cells, HIV-1 particles captured by DCs are passed directly to T cells. Moreover, together these studies indicate that while CD169^+^ macrophages in general have an important role in preventing systemic dissemination of viral infections, viruses have hijacked this receptor to mediate trans-infection of immune cells.

#### 4.1.2. Bacterial Infection

CD169^+^ macrophages are involved in bacterial infections with sialylated lipopolysaccharides, lipooligosaccharides, or capsular polysaccharides, including *Group B Streptococcus* (*GBS*), *Streptococcus pneumoniae*, and *Campylobacter jejuni* [11]. *GBS* is a pathogenic bacterium that can trigger severe infections in neonates and the elderly and can bind multiple Siglec family members [129,130]. CD169-mediated phagocytosis is important for its clearance. A recent study found that newborns have significantly lower expression levels of CD169 in lung alveolar macrophages at birth compared to the postnatal period. This reduced CD169 expression in newborns was specifically linked to increased susceptibility to *GBS* infection [131]. Another study showed intracellular replication of *S. pneumoniae* in splenic CD169^+^ macrophages that is followed by bacteria release into the circulation [132]. *C. jejuni* strains containing α2,3-sialic acid lipooligosaccharides that mimic gangliosides can bind to CD169^+^ myeloid cells and this may contribute to the development of Guillain-Barré syndrome [133,134,135,136,137,138]. In addition to *GBS* and *C. jejuni*, the binding of CD169 to *Neisseria meningitidis* has also been described [136,139,140].

#### 4.1.3. Parasitic and Fungal Infection

Limited studies have pointed to a role for CD169^+^ macrophages and the CD169 receptor in parasitic and fungal infections. Similar to their role in viral and bacterial infections, CD169^+^ macrophages can control parasites and fungi through indirect mechanisms, as has been shown for Plasmodium and Candida albicans infection [30,31,110,141,142,143]. Only for Leishmania a clear interaction between sialic acids displayed by the parasite and the CD169 receptor has been described. CD169^+^ macrophages bind Leishmania promastigotes in a CD169-dependent manner, and blocking CD169 decreases infection [144,145,146]. Interestingly, one study pointed to the binding of CD169 to Trypanosoma cruzi, which is also decorated with sialic acids, but this has not been further explored [147].

### 4.2. The Role of CD169-Expressing Cell Subsets in Autoimmunity

CD169^+^ monocytes and macrophages have been implicated in multiple inflammatory disorders. In rheumatoid arthritis, a chronic autoinflammatory condition that affects the joints, circulating CD169^+^ monocytes and joint-resident CD169^+^ macrophages are increased and monocyte presence correlates with a worse prognosis [4,148,149,150]. After anti-rheumatic therapy, CD169^+^ monocyte frequencies decrease [148]. CD169 expression on circulating monocytes is elevated in systemic lupus erythematosus (SLE), a chronic multi-organ autoimmune disease [151,152,153,154,155]. This increase is linked to the type I IFN signature that contributes to SLE progression. Consequently, CD169^+^ monocytes also strongly correlate with disease severity [151,156]. CD169 can also be detected in its soluble form in the blood which facilitates its use as a biomarker for disease progression [153]. CD169^+^ monocytes have additionally been linked to disease activity in dermatomyositis, systemic sclerosis, and Sjögren’s syndrome [5,157,158]. Multiple sclerosis (MS) is characterized as a myelin-targeting autoimmune disorder resulting in brain lesions. CD169 expression is increased in these MS lesions [159,160]. One of the mechanisms by which CD169^+^ macrophages may propagate MS is by binding and inhibiting Tregs [19]. Conflicting results regarding increased expression of CD169 on circulating monocytes in MS patients are reported, potentially due to heterogeneity in the study populations [159,160,161]. As previously mentioned, CD169^+^ macrophages are present in the gut lamina propria. In a mouse model of colitis, CD169^+^ macrophages are increased and they respond to mucosal damage by producing CCL8 which recruits inflammatory monocytes and propagates disease [104,106,162]. Together, these studies indicate that in a variety of autoimmune diseases, CD169 is upregulated on circulating monocytes and inflammatory macrophages.

### 4.3. The Role of CD169-Expressing Cell Subsets in Cancer

The presence of CD169^+^ macrophages in the tumor-draining lymph nodes (TDLNs) as well as within the tumor microenvironment, so-called CD169^+^ tumor-associated macrophages (TAMs) and their potential role in tumor development and tumor-specific immune responses have been investigated for a multitude of cancers, but represent a yet incompletely understood aspect of cancer immunology.

A higher density of CD169⁺ macrophages in TDLNs, associated with increased CD8⁺ cytotoxic T-cell infiltration in primary tumor sites, was identified as an independent prognostic factor for improved overall survival in melanoma patients [163]. In both orthotopic mouse melanoma models and clinical specimens from melanoma patients, CD169^+^ SSMs in TDLNs exhibited metastasis-suppressive functions through active scavenging of tumor-derived extracellular vesicles [164]. CD169^+^ SSMs in TDLNs in breast cancer patients are associated with favorable clinical features and improved prognosis [165,166]. A mouse model for breast cancer revealed that the anti-metastatic function of these CD169^+^ macrophages in TDLNs required functional crosstalk with B cells [167]. A significant positive correlation exists between CD169 expression in TDLN macrophages and CD8^+^ T cell infiltration in esophageal cancer, gastric cancer, and colorectal cancer along with improved overall survival and enhanced anti-cancer immune responses [168,169,170,171]. In bladder cancer patients, the presence of CD169^+^ macrophages in TDLNs was positively associated with improved cancer prognosis [172,173]. Low CD169 expression in TDLNs was significantly associated with an increased risk of prostate cancer death in mouse and human data [174]. Mouse model studies showed that the success of anti-PD-L1 immunotherapy for cancer was dependent on the presence of CD169^+^ macrophages in the TDLNs [175]. In conclusion, multiple studies across different tumor types have demonstrated that higher levels of CD169-expressing macrophages in TDLNs correlate positively with better disease prognosis, highlighting the potential significance of these specialized macrophages in anti-tumor responses.

In a number of cancer types, the presence of CD169^+^ TAMs tends to correlate with better patient prognosis. This has been shown for glioblastoma, hepatocellular carcinoma (HCC), pancreatic cancer, and gastric cancer [176,177,178,179,180,181]. A low dose of type I IFNs was proposed to induce CD169^+^ macrophage polarization and to enhance CD8^+^ T cell activation in HCC [182].

In contrast, transcriptional profiling of TAMs in breast and endometrial cancers reveals that CD169 serves as a specific TAM marker strongly associated with clinical aggression [183]. CD169^+^ TAMs in primary breast tumors are spatially associated with tertiary lymphoid-like structures and correlate with regulatory T and B cells, leading to a worse prognosis in advanced breast cancer [184]. CD169^+^ TAMs might inhibit CD8^+^ T cell anti-tumor effects through PD-L1 upregulation in the tumor microenvironment of breast cancer and lung metastases [185,186]. Similarly, in colorectal cancer patients, significantly increased percentages of CD14^+^ CD169^+^ cells were found in both circulating monocytes and TAMs compared to healthy controls. These increased levels correlated positively with the advanced disease stage [187]. CD169^+^ TAMs correlated with worse overall survival in bladder cancer [172]. These findings suggest that the tumor microenvironment plays a crucial role in influencing CD169^+^ macrophage function. CD169^+^ TAMs exhibit dual roles in cancer progression, correlating with improved prognosis in glioblastomas, HCC, and gastric cancers through immune activation, while promoting aggression in breast, endometrial, and bladder cancers via immunosuppressive mechanisms. This context-dependent duality underscores the critical influence of the tumor microenvironment on CD169^+^ TAM function, necessitating further research to unravel tissue-specific mechanisms and therapeutic opportunities.

## 5. CD169-Targeting Delivery Systems

Due to the strategic position of CD169^+^ macrophages in the spleen and lymph nodes and their capacity to stimulate B and T cell responses, an interesting strategy to enhance vaccination efficacy is to include molecules that enhance capture by these cells. Different approaches that have been tested include CD169-binding antibodies or ligands and have mostly been investigated in the context of cancer to stimulate T cell responses, while vaccines for infectious diseases are under-explored. The different platforms currently under investigation as CD169-focused vaccination strategy are described below (Figure 2).

This figure depicts direct interactions of CD169 with various pathogens (left), as well as different formats of targeting moieties specific to CD169 (right). The bottom panel illustrates the interactions of CD169^+^ macrophages with other immune cells. Created in BioRender. den Haan, J. (2025) https://BioRender.com/t39k632 (accessed on 19 February 2025).

### 5.1. CD169-Targeting Antibodies as a Cancer Vaccination Platform

Studies that used antibodies specific for CD169 to enable antigen targeting to splenic CD169^+^ macrophages not only elucidated the transfer of antigen to dendritic cells but also demonstrated similar T cell priming efficacy when compared to antigen targeting to DCs [16,63,188]. In these studies, antigens were conjugated to antibodies specific for CD169 and injected together with an adjuvant. This antigen targeting strongly increased CD8^+^ and CD4^+^ T cell priming and revealed that antigens were transferred preferentially to cDC1s, but also to a lower extent to cDC2s. The transfer was dependent on CD169 itself and the subsequent T cell priming partially on DNGR-1 [16]. Targeting antigens to splenic CD169⁺ macrophages also stimulated strong, high-affinity antibody responses and promoted germinal center B cell activity through effective T follicular helper cell differentiation [51]. These data underscore the role of CD169^+^ macrophages in the activation of adaptive immunity.

Since these initial studies showed the remarkable potential of CD169-targeted vaccination strategies to induce CD8^+^ T cell priming, this was further explored by subsequent studies. The conjugation of both immunogenic peptide and protein efficiently stimulated effector and memory CD8^+^ T cell responses with anti-CD169 antibodies that were comparable to those induced with DEC205-specific antibodies [188]. Moreover, the CD169 peptide and protein conjugates stimulated potent anti-tumor reactivity [188]. By conjugating T cell epitopes from melanoma-associated antigens such as Trp2, gp100, and MART-1 to antibodies specific for mouse and human CD169, strong T cell responses were obtained in both mouse models and human cell studies, suggesting broad translational potential [189,190].

Next, a site-specific conjugation technique was explored for the ligation of peptide antigens to CD169-specific antibodies. By utilizing CRISPR/HDR engineering, the researchers created chimeric antibodies with a Sortase A recognition motif and a SpyTag, enabling site-specific attachment of peptide antigens for personalized neoantigen vaccines using proximity-based Sortase A-mediated ligation. The study demonstrated that these antibody-neoantigen conjugates effectively stimulated robust T cell immune responses when targeted to CD169^+^ macrophages [191]. These studies combined indicate that antigen targeting to CD169 via antibodies generates strong, specific T cell responses that can potentially be used to enhance the effectiveness of existing immunotherapies.

### 5.2. CD169-Targeting Nanoparticles as Cancer Vaccination Strategy

Next to antibodies, nanoparticles that contain CD169-binding ligands have also been investigated extensively. Nycholat et al. and Chen et al. synthesized synthetic ligands with high affinity for CD169 and linked these to PEGylated lipids for incorporation into liposomes [192,193]. The liposomes were efficiently taken up by CD169^+^ macrophages in vitro and in vivo and after incorporation of the model antigen ovalbumin showed enhanced ability to activate CD8^+^ and CD4^+^ T cells [193,194].

Liposomes containing gangliosides as natural ligands for CD169 were additionally studied. Although the affinity of gangliosides for the CD169 receptor is lower compared to antibodies, liposomes incorporating CD169-binding gangliosides were efficiently taken up by human and mouse CD169^+^ antigen-presenting cells (APCs) [62,195,196]. In these studies, 3 mol% ganglioside GM3 was incorporated in liposomes without the addition of PEG as it shields GM3 on the surface and blocks its ability to bind CD169 [195]. Experiments with liposomes containing different gangliosides, including GM3, and loaded with cancer antigens resulted in increased antigen presentation by human IFNα-treated monocyte-derived DCs, monocytes, or enriched blood CD169^+^ Axl^+^ Siglec-6^+^ DCs to antigen-specific T cell clones [190,196]. Intravenous injection of ovalbumin-loaded GM3 liposomes with adjuvant in vivo led to efficient uptake by splenic CD169^+^ macrophages and induced antigen-specific T cell and germinal center B cell responses. The CD8^+^ T cell responses were dependent on cDC1s [62,195,197]. Interestingly, platelets appeared to bind to liposomes and to CD169^+^ macrophages, and depletion of platelets significantly diminished the antigen-specific T cell response to antigen-loaded GM3 liposomes [197].

CD169-targeting liposomes can also be harnessed with natural killer T (NKT) cell-activating α-Galactosyl ceramide (αGC). CD169^+^ macrophages activate NKT cells through CD1d-mediated presentation of αGC [60,198,199,200] and these innate cells subsequently help the maturation of cDC1s and cDC2s [199]. Interestingly, the depletion of macrophages gives opposing results in different studies. Kawasaki et al. showed that the depletion of macrophages with clodronate liposomes completely aborted the activation of NKT cells after treatment with CD169-targeting αGC liposomes [198]. In contrast, Grabowska et al. reported no effect of diphtheria toxin-induced removal of CD169^+^ macrophages on NKT cell activation with GM3 αGC liposomes [199]. This could be due to the different depletion methods used since the clodronate liposome method also depletes other APCs, such as DCs, that could aid in NKT cell activation [63]. Ovalbumin-loaded, GM3 αGC liposomes induced a strong CD8^+^ T cell response in vivo. In comparison, ovalbumin-loaded, GM3 liposomes co-injected in vivo with anti-CD40 and poly I:C led to a combined CD8^+^ and CD4^+^ T cell response [199].

To ensure the immunogenicity of liposomal antigens, incorporating the right adjuvant is of importance. Interestingly, the addition of a TLR7 agonist to CD169-targeting liposomes injected in vivo resulted in a predominant CD8^+^ T cell-mediated immune response, whereas injection of CD169-targeting, antigen-loaded liposomes in the absence of TLR7 agonist led to CD4^+^ T cell activation [194]. Nijen Twilhaar et al. investigated the effect of incorporation of TLR4 or TLR7/8 ligands combined with inflammasome stimulants 1-palmitoyl-2-glutaroyl-sn-glycero-3-phosphocholine (PGPC) or muramyl dipeptide (MDP) in GM3 liposomes. Overall, both TLR ligands increased DC maturation and PGPC modestly enhanced this. However, the antigen-specific response to ovalbumin-loaded GM3 liposomes was only increased when a TLR4 ligand, but not the TLR7/8 ligand or any inflammasome stimulant was added [201]. It should be noted that nanoparticle-incorporated adjuvants administered in vivo are efficiently taken up by CD169^+^ macrophages in the draining lymph nodes, even without CD169 targeting. Polymer-induced co-assembly of iron oxide nanoparticles, a STING agonist, and ovalbumin resulted in a vaccination strategy that induced a type I IFN response and antigen-specific T cell increase after subcutaneous injection in vivo [202]. Moreover, a liposomal saponin-based adjuvant called QS-21 strongly increased the antigen-specific immune response after intramuscular immunization in vivo [203]. Although both strategies are untargeted, liposomal clodronate depletion of CD169^+^ macrophages revealed these macrophages were essential for the adjuvant effect. Potentially, targeting these vaccination strategies to CD169 could enhance their efficacy even further.

In addition to the incorporation of natural ligands, antibodies or related molecules can also be attached to nanoparticles as a vaccination strategy. This way, specific receptors on antigen-presenting cells (APCs) can be selected for targeting and different affinities can be researched. However, attaching antibodies to nanoparticles results in increased nanoparticle size and poor stability [204]. To overcome these challenges, multiple groups [205,206,207] investigated the conjugation of single-domain antibodies, also known as nanobodies, to liposomes. Initially, research mainly focused on the attachment of nanobodies that directly target tumor-associated markers such as EGFR or HER2 to liposomes loaded with a therapeutic agent [205,206,207]. More recently, immune cell-targeting nanobodies have also been attached to liposomes. Bouma et al. produced nanobody liposomes specific for CD169 and DC-specific intercellular adhesion molecule-3-grabbing non-integrin (DC-SIGN) expressed on APCs. The CD169-specific nanobodies were attached to PEGylated liposomes through a maleimide-cysteine interaction and strongly bound to mouse and human CD169^+^ APCs. Human CD169^+^ monocyte-derived DCs presented the incorporated cancer antigens from the CD169 nanobody liposomes to human CD8^+^ T cells. Moreover, intravenously injected ovalbumin-loaded liposomes efficiently bound to CD169^+^ macrophages in the spleen and enhanced antigen-specific T cell responses after 7 days [208]. These combined studies indicate that nanoparticles that bind to CD169 effectively activate T cell responses and can be used to enhance anti-cancer T cell responses.

### 5.3. CD169 Targeting in Viral Infections

Apart from improving vaccination strategies, CD169 targeting has been studied extensively to gain a better understanding of HIV dissemination. HIV can bind DCs through the interaction of viral membrane gangliosides GM1 and GM3 with CD169 [116,117]. Artificial virus nanoparticles (AVNs) were designed to mimic the viral membrane and contained a solid metal core to enable advanced imaging techniques and GM3 molecules in their lipid layer [209,210,211]. After binding to CD169, these GM3-containing AVNs accumulated in non-lysosomal compartments similar to the virus-containing compartment. This compartment plays a crucial role in viral dissemination by facilitating the transfer of viral particles from DCs to T cells. Interestingly, while GM3-containing AVNs routed to the virus-containing compartment, this was not observed with GM1-containing AVNs or GM3-containing liposomes [212]. Experiments with alternative nanoparticle designs using solid polymer cores made from either PLA (polylactic acid) or PLGA (poly(lactic-co-glycolic acid)) also demonstrated that only the stiff GM3-containing PLA AVNs but not the softer PLGA AVNs or liposomes localize in the virus-containing compartment and avoid the lysosomal pathway [213]. GM3 AVNs were also considered as a strategy to enhance the effectivity of chimeric antigen receptor (CAR) T cells. PLA core GM3 nanoparticles were synthesized with FITC to activate anti-FITC CAR T cells. The nanoparticles were bound by CD169-expressing APCs but were still accessible for CAR T cells, which were subsequently activated by the FITC GM3 particles [214].

The success of antiretroviral therapies against HIV is limited by poor penetration of antiretrovirals and deficient immunosurveillance which results in residual virus replication. This could be overcome by targeting antiretrovirals to potential virus reservoir sites. Zang et al. successfully produced lipid-coated mesoporous silica nanoparticles containing GM3 as a delivery system for HIV antiretroviral drugs specifically targeting CD169^+^ macrophages [215]. Similarly, GM3 PLA nanoparticles loaded with antiretrovirals successfully inhibited HIV-1 replication in CD169^+^ macrophages [216]. A different antiviral therapeutic strategy involves blocking CD169 to inhibit CD169-mediated viral spread. Monoclonal antibodies specific for CD169 effectively inhibited binding and cytoplasmic entry of HIV-I and Ebola virus in LPS- or IFNα-treated DCs [118]. Three of these antibodies were subsequently humanized in an effort to move this therapeutic approach closer to clinical testing [217].

In conclusion, these studies show that GM3 binding to CD169 delivers viruses and particles to virus-containing compartments and indicate that blocking this interaction by antibodies and the targeting of antiretroviral drugs to CD169 may be potential strategies to inhibit virus dissemination.

## 6. Conclusions and Future Outlook

This review outlined CD169^+^ macrophage tissue distribution and functions in homeostasis, infectious diseases, cancer, and autoimmunity. Additionally, the different approaches to delivering antigens or antiviral drugs to CD169^+^ macrophages were discussed. Although the discussed studies have provided many interesting insights about this macrophage subtype, several questions about its function remain.

CD169^+^ macrophages in secondary lymphoid organs have important, well-described functions in bridging innate and adaptive immunity through the capture of pathogens and antigen transfer to DCs and B cells. Resident CD169^+^ macrophages in other tissues, such as Kupffer cells and alveolar macrophages, have a variety of roles that also include scavenging pathogens. Whether these macrophages can transfer antigens to other DCs or B cells similar to splenic and lymph node CD169^+^ macrophages is still unclear. Next to CD169 expression in a steady state, CD169 is also upregulated on monocytes and macrophages under inflammatory conditions and in a number of cancers. Upregulation of CD169 in several inflammatory diseases is linked to a type I IFN signature and can be used as a marker for disease severity and therapy response, for example in SLE. Does CD169 have similar or different functions on these inflammatory macrophages as compared to the tissue-resident CD169^+^ macrophages? In MS and colitis, studies suggest that CD169^+^ macrophages propagate inflammation. In other inflammatory diseases, it is not yet clear whether these macrophages contribute to disease progression or are solely present due to the large amounts of type I IFNs released. Further elucidating the role of CD169 on inflammatory monocytes and macrophages could provide important leads for future treatments in several disease settings.

Novel vaccination strategies are being developed that exploit the efficient manner in which CD169^+^ macrophages in the spleen and lymph nodes instigate an adaptive immune response. CD169-binding antibody- or nanoparticle-based methods can successfully increase the number of antigen-specific B and T cells. So far, most studies use these methods to generate anti-cancer immune responses, but pathogen antigen-containing nanoparticles that bind to CD169 could also be successful in providing increased protection against infection. Although a traditional vaccination approach is unlikely to be of use as a treatment for autoimmune diseases, tolerizing vaccines that target CD169 could potentially be harnessed to diminish the immune response instead.

To conclude, CD169 is an important molecule on several tissue-resident macrophage subsets that is involved in many immune processes. Future studies are expected to reveal more insights about this molecule and the role of CD169^+^ macrophages in multiple disease settings and how CD169-targeting strategies can be helpful in resolving these diseases.

## Figures and Tables

**Figure 1 vaccines-13-00330-f001:**
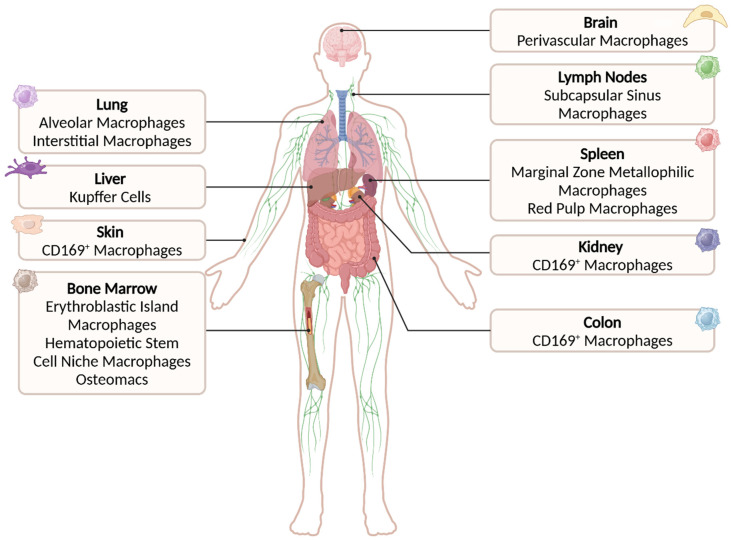
CD169-expressing macrophage subsets in different organs.

**Figure 2 vaccines-13-00330-f002:**
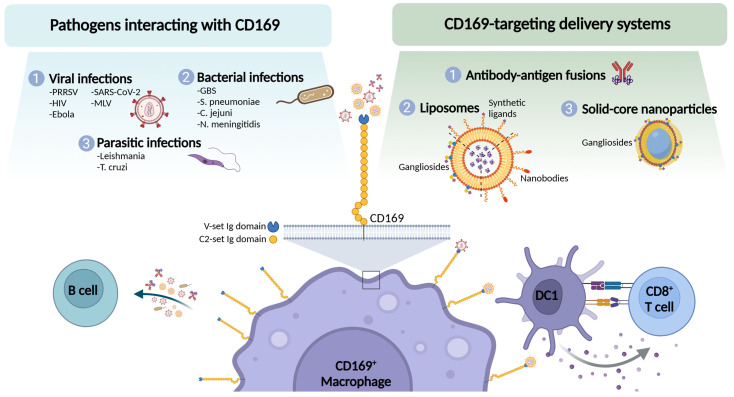
An overview of the interaction of pathogens and vaccination strategies with CD169.

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
