# Peer review of "Exploring CD169+ Macrophages as Key Targets for Vaccination and Therapeutic Interventions"

_vaccines, 2025, doi:10.3390/vaccines13030330_

Round 1
Reviewer 1 Report
Comments and Suggestions for Authors
The manuscript by Bian et al. outlines the role of CD169 molecule and CD169+ macrophages in immunology as well as in application to vaccination, including tumor immunotherapy. Overall, the description of CD169 macrophage functions is quite extensive and well-researched, being an excellent comprehensive review of the cells.
Specific comments
- The authors should provide a table that summarizes Section 3 by focusing on the phenotype and function of CD169-expressing macrophages or monocytes in each organ (and each species, if possible) (e.g., Table 2 in the review article in Immunity [Hashimoto et al., Immunity 35: 323-335, 2011, PMID 21943488]; Table 1 in the review article in Immunol Review [Halder and Murphy, Immunol Rev 62:25-35, 2014, PMID 25319325]). This information would provide comprehensive information explaining the phenotypical and functional heterogeneity of CD169+ macrophage across organs.
- The first line of Section 3.1, “CD169 was originally discovered on ….” explains redundantly with the first line of Section 2, “CD169 was first discovered as…”.
- The second line of Section 3.2: please spell out MMMs and SSMs.
- The seventh line of Section 3.2: “MMMS” should be changed to “MMMs”.
- The structure of each CD169 liposome mentioned in Section 5.2 should be separately shown in Figure 2.
Reviewer 2 Report
Comments and Suggestions for Authors
This review introduces the homeostatic functions, tissue-specific distribution, and roles of CD169+ macrophages in various pathological conditions (including infectious diseases, inflammation, and cancer), and presents multiple vaccine and therapeutic intervention strategies targeting CD169.
There are some problems, which must be solved before it is considered for publication.
1. To enhance the readability of the paper, I suggest presenting the expression of CD169 in various tissues in a tabular format, indicating the tissue, expressing cells, and the specific pathways and functions it exerts.
2. Improve the content of your Fig2. In the section on CD169-targeting delivery systems, add specific regulatory factors and signaling pathways, such as the fact that CD169 antigen-antibody conjugation enhances T cell activation primarily through cDC1.
3. In the conclusion and outlook, the discussion on the roles of CD169 macrophages in different inflammatory conditions is not comprehensive enough.
The English could be improved to more clearly express the research.
